# Multimodality Cardiovascular Imaging of Cardiotoxicity Due to Cancer Therapy

**DOI:** 10.3390/life13102103

**Published:** 2023-10-23

**Authors:** Carla Contaldi, Vincenzo Montesarchio, Dario Catapano, Luigi Falco, Francesca Caputo, Carmine D’Aniello, Daniele Masarone, Giuseppe Pacileo

**Affiliations:** 1Heart Failure Unit, Department of Cardiology, AORN dei Colli-Monaldi Hospital, 80131 Naples, Italy; dariocat90@gmail.com (D.C.); luigifalco94@libero.it (L.F.); danielemasarone@ospedalideicolli.it (D.M.); giuseppe.pacileo@ospedalideicolli.it (G.P.); 2Division of Medical Oncology, AORN dei Colli-Monaldi Hospital, 80131 Naples, Italy; vincenzo.montesarchio@ospedalideicolli.it (V.M.); francesca.caputo@ospedalideicolli.it (F.C.); carmine.daniello@ospedalideicolli.it (C.D.)

**Keywords:** cardiotoxicity, echocardiography, cardiac magnetic resonance, cardiac computed tomography, nuclear imaging, cancer therapy

## Abstract

Cancer therapies have revolutionized patient survival rates, yet they come with the risk of cardiotoxicity, necessitating effective monitoring and management. The existing guidelines offer a limited empirical basis for practical approaches in various clinical scenarios. This article explores the intricate relationship between cancer therapy and the cardiovascular system, highlighting the role of advanced multimodality imaging in monitoring patients before, during, and after cancer treatment. This review outlines the cardiovascular effects of different cancer therapy classes, offering a comprehensive understanding of their dose- and time-dependent impacts. This paper delves into diverse imaging modalities such as echocardiography, cardiac magnetic resonance imaging, cardiac computed tomography, and nuclear imaging, detailing their strengths and limitations in various conditions due to cancer treatment, such as cardiac dysfunction, myocarditis, coronary artery disease, Takotsubo cardiomyopathy, pulmonary hypertension, arterial hypertension, valvular heart diseases, and heart failure with preserved ejection fraction. Moreover, it underscores the significance of long-term follow-up for cancer survivors and discusses future directions.

## 1. Introduction

The evolving landscape of cancer treatment has brought about significant advancements, allowing for improved patient survival rates. However, the cardiovascular impact of many cancer therapies is now better understood and has been associated with a range of cardiotoxic effects, highlighting the pressing need for effective monitoring and management [1,2,3,4].

Despite the existence of several international guidelines and position papers regarding the cardiovascular monitoring and management of cancer drug recipients, their limited empirical basis makes it challenging to determine a practical approach suitable for every clinical scenario [5,6,7,8,9].

This article delves deep into the complex interplay between cancer therapy and the cardiovascular system, emphasizing the value of a state-of-the-art multimodality imaging approach to stratify the patient’s risk before starting cancer therapy, identify early cardiovascular injury during treatment and detect cardiovascular injury in long-term.

Beginning with a detailed exploration of the newly introduced terminology from the 2022 European Society of Cardiology (ESC) guidelines [9] and American College of Cardiology Cardio-Oncology and Cardiovascular Imaging Leadership Councils [10], this review elaborates on the distinct classifications of cardiovascular toxicity due to cancer therapy.

The potential cardiovascular side effects of different classes of cancer therapies are elucidated, offering a comprehensive understanding of their dose- and time-dependent impacts.

A pivotal section of this review emphasizes the diverse imaging modalities available for detecting cardiotoxicity. The roles of echocardiography, cardiac magnetic resonance imaging (CMR), cardiac computed tomography (CCT), and nuclear imaging techniques are explored, highlighting their strengths and potential limitations.

The significance of each modality is further delineated across various conditions associated with cancer therapy, such as cardiac dysfunction, myocarditis, coronary artery disease (CAD), Takotsubo cardiomyopathy, pulmonary hypertension, arterial hypertension, valvular heart diseases, and heart failure (HF) with preserved ejection fraction (EF).

Lastly, this review underscores the significance of long-term follow-up for cancer treatment survivors and hints at the promising future directions in the field, including the exploration of novel imaging markers like the PET/CT scan.

## 2. Cardiotoxicity, Risk Stratification, Stage of Evaluation

The 2022 ESC guidelines [9] introduced the comprehensive term “cancer therapy-related cardiovascular toxicity (CTR-CVT)” to distinguish it from “cancer therapy-related cardiac dysfunction (CTRCD)”, which encompasses cardiac injury, cardiomyopathy, and HF. CTRCD is further categorized into symptomatic and asymptomatic, with the latter defined by an EF threshold of 50% along with a new relative decline in global longitudinal strain (GLS) of at least 15% from baseline and/or an increase in cardiac biomarkers.

The American College of Cardiology Cardio-Oncology and Cardiovascular Imaging Leadership Councils have presented an imaging definition of CTRCD that complements the clinical classifications of cardiotoxicity as described by the International Cardio-Oncology Society (ICOS) [10]. These include “Definite CTRCD”, characterized by a reduction in left ventricle (LV) EF of ≥10% to a value below 50%, and “Possible CTRCD”, characterized by a reduction in LVEF of ≥10% to a value between 50 and 55%, a reduction in LVEF by <10 percentage points to a value below 50%, or a relative reduction in GLS of ≥15% without a significant reduction in LVEF [11].

As previously outlined in the 2020 Heart Failure Association (HFA)/ICOS position paper [8] and later endorsed by the 2022 ESC guidelines [9], risk stratification for cardiovascular toxicity (preferably employing HFA-ICOS risk assessment) before initiating potentially cardiotoxic cancer therapy is recommended for all patients. Additionally, it is advised to engage in a multidisciplinary discussion of the risk–benefit balance of cardiotoxic anticancer treatment in high- and very high-risk patients before commencing treatment to minimize unnecessary interruptions in antineoplastic therapy [9].

Patients who do not experience disease progression or have a poor prognosis in the first year following chemotherapy discontinuation are classified as long-term survivors and are assigned to a specific risk and surveillance category [9].

Therefore, it is recommended to assess the cardiovascular status of cancer patients before, during, and after receiving cardiotoxic treatment. The frequency of cardiovascular monitoring should be tailored to the type of drug, treatment settings, combination with other therapies, and, most importantly, the patient’s risk of cardiotoxicity, which is a dynamic parameter that should be reassessed over time [9].

For cardioprotection, the 2022 ESC guidelines suggest the use of ACE inhibitors (ACEI) or angiotensin receptor blockers (ARBs), betablockers, and statins for primary prevention in patients at high and very high risk (class IIa) [9].

Currently, sodium–glucose cotransporter type 2 inhibitors (SGLT2i) are emerging as primary therapy for both HF with reduced EF (HFrEF) and HF with preserved EF (HFpEF) due to their multiple cardioprotective effects [12,13]. Chemotherapeutic agents can induce cardiac toxicity through various mechanisms, including oxidative stress, mitochondrial dysfunction, and interference with cardiac cell signaling pathways [12,13].

In mouse models, SGLT2i can prevent anthracycline-induced LV dysfunction through several mechanisms, including increased cardiac energy production, reduced oxidative stress, preservation of mitochondrial function, anti-inflammatory effects, and decreased cell death and myocardial fibrosis [14]. In addition, SGLT2i have been shown to reduce the rate of HF-related hospitalizations following anthracycline chemotherapy [15].

Incorporating SGLT2i into the management of cancer patients receiving cardiotoxic chemotherapy could be a crucial strategy for mitigating chemotherapy-related cardiac complications. Nevertheless, their precise role in clinical practice necessitates further investigation and randomized controlled clinical trials involving human patients undergoing cancer treatments.

## 3. Classes of Cancer Therapy and Their Cardiovascular Effects

Anthracyclines (doxorubicin, epirubicin, idarubicin, daunorubicin, mitoxantrone), some of the oldest cancer drugs, are still used for various conditions like lymphoma, leukemia, osteosarcoma, and breast cancer. They work against cancer by disrupting DNA and RNA synthesis, generating free radicals that harm DNA, inhibiting topoisomerase II (Top2), and altering histones. This toxicity is linked to their anticancer effects, including Top2 inhibition and reactive oxygen species (ROS) production [2,16].

Anthracycline-related toxicity can be categorized as acute (within two weeks of treatment), early (within a year after therapy completion), and late (over a year after therapy). Cardiotoxicity can be dose-dependent (early and late) and reversible (acute) or challenging to reverse (early and late).

Risk factors for anthracycline-induced side effects include being female, having pre-existing conditions (coronary artery disease, diabetes, hypertension, renal failure), extremes of age (pediatric or elderly), combined therapy with microtubule inhibitors, and concomitant heart-related radiotherapy [2]. Anthracycline-induced CTRCD is a cumulative and dose-dependent process with variable onset, which may or may not present with symptoms. A planned cumulative doxorubicin dose of ≥250 mg/m^2^ or equivalent is considered high risk. In adults at high and very high cardiovascular toxicity risk, liposomal anthracyclines should be considered [9].

HER2-targeted monoclonal antibodies (trastuzumab, pertuzumab, trastuzumab emtansine [T-DM1], trastuzumab-deruxtecan) play a crucial role in treating HER2-positive breast cancer, both in early and metastatic stages. Trastuzumab can also be used for patients with HER2-overexpressing metastatic gastric adenocarcinomas in combination with platinum-based chemotherapy and either capecitabine or 5-fluorouracil (5-FU).

These antibodies block the HER-2/neureceptor, halting the activation of intracellular growth factors.

Cardiac dysfunction arises from the disruption in signaling between the HER-2/ERBB2 receptor and the neuregulin ligand, a pathway responsible for growth, repair, and homeostasis.

Cardiotoxicity typically emerges during therapy, is reversible, and can lead to left ventricular dysfunction in 15–20% of patients. Factors like simultaneous therapy with anthracyclines, age over 50, obesity, LVEF < 50%, and prior surgery increase the risk of cardiotoxicity [2,9].

Fluoropyrimidines (antimetabolites 5-FU and its oral prodrug Capecitabine) are primarily used for treating gastrointestinal cancers and advanced breast cancer. The most common CTR-CVT associated with these drugs include angina pectoris, ECG abnormalities related to ischemia, arterial hypertension, Takotsubo syndrome, and myocardial infarct, even in patients with normal coronary arteries. Less common CTR-CVT events involve myocarditis, arrhythmias, and damage to peripheral arteries like Raynaud’s phenomenon and ischemic stroke.

The incidence of myocardial ischemia can reach up to 10%. Various mechanisms, including coronary vasospasm and endothelial injury, contribute to 5-FU-induced myocardial ischemia. The risk of CTR-CVT significantly increases in cancer patients with pre-existing CAD [9].

Vascular endothelial growth factor inhibitors (VEGF Inibitors) are available either as monoclonal antibodies (Afibercept, Bevacizumab, Ramucirumab), administered intravenously and targeting circulating VEGF, or as small-molecule tyrosine kinase inhibitors (TKIs: Sorafenib, Sutinib, Vandetanib, Pazopanib) taken orally, targeting VEGF receptors.

An abnormal VEGF signaling pathway appears to play a crucial role in the development of various cancer types. VEGF inhibitors are typically employed in the treatment of renal, thyroid, and hepatocellular carcinomas.

Hypertension is the most common cardiac side effect (with an incidence greater than 10%), and it is a dose-dependent class effect that can usually be reversed by discontinuing VEGF inhibitors. The risk is higher in patients with pre-existing hypertension, prior anthracycline treatment, advanced age, a history of smoking, hyperlipidemia, and/or obesity.

Other common cardiovascular complications may include HF/LV dysfunction, QTc interval prolongation, and acute vascular events such as aortic dissection, stroke, arterial thrombosis, acute coronary events, and vasospasm [9,11].

Tyrosine kinase inhibitors (TKIs) targeting BCR-ABL include Imatinib (1st gen)-Nilotinib (2nd gen)-Dasatinib (2nd gen)-Bosutinib (2nd gen)-Ponatinib (3rd gen).

Chronic myeloid leukemia results from the abnormal activation of ABL1 kinase due to a chromosomal translocation. Small-molecule TKIs targeting BCR-ABL have been proven to be effective in chronic myeloid leukemia treatment. The toxicities associated with these TKIs are distinct and result from the ‘off-target’ effects of each drug. Dasatinib is linked to group 1 pulmonary hypertension (PAH), HF, and pleural and pericardial effusion. Nilotinib and Ponatinib are generally associated with vascular events. Second-generation BCR-ABL TKIs may induce QTc interval prolongation.

The risk of cardiovascular toxicity is elevated in patients aged 65 years and those with underlying diabetes mellitus, hypertension, or pre-existing CAD [9,11].

Bruton TKIs (Ibrutinib, Acalabrutinib) are used to treat lymphoid malignancies. Bruton TKIs play a crucial role in B-cell development. Ibrutinib, a first-in-class irreversible oral inhibitor of BTK, has proven to be highly effective in chronic lymphocytic leukemia and related B-cell malignancies.

These disorders are usually diagnosed in elderly patients with more comorbidities. Ibrutinib has been associated with bleeding diathesis, infections and increased risk of arterial hypertension, atrial fibrillation and HF [9,11].

Proteosome inhibitors (Carfilzomib, Bortezomib, Ixazomib) are commonly employed in the treatment of multiple myeloma and mantle cell lymphoma.

The proteasome’s function is to break down abnormal proteins tagged with ubiquitin. In cancer cells, this pathway is overactive. The anticancer mechanism of this group involves inhibiting the proteasome by binding to one of its subunits.

The suggested mechanism of cardiotoxicity includes oxidative stress on cardiomyocytes leading to apoptosis or temporary endothelial dysfunction.

Cardiovascular adverse events may encompass arterial hypertension, HF, acute coronary syndromes, arrhythmias, pulmonary hypertension, and thrombotic events. HFpEF can have a significant adverse effect, especially with carfilzomib. Pre-existing cardiovascular disease and concurrent administration of anthracyclines are risk factors for cardiac dysfunction [2,9].

Immune checkpoint inhibitors (ICI) are typically utilized in the treatment of advanced melanoma, metastatic kidney cancer, and non-small cell lung cancer. These therapies target immune checkpoints, which are proteins expressed in T cells that act to suppress their activation when they come into contact with healthy cells.

ICIs are monoclonal antibodies that interfere with immune regulators, including cytotoxic T-lymphocyte-associated antigen-4 (CTLA-4) (such as Ipilimumab and Tremelimumab), programmed death-1 (PD-1) (such as Nivolumab, Cemiplimab, and Pembrolizumab), and programmed death-ligand 1 (PD-L1) (such as Atezolizumab, Avelumab, and Durvalumab), found in cancer cells. This interference leads to an enhanced immune response, allowing T cells to attack and destroy cancer cells by preventing these checkpoints from binding with their partner proteins and inhibiting the ‘off’ signal. However, ICI treatment can also result in excessive T-cell activation against normal tissues, leading to immune-related adverse events.

Immune-related cardiovascular complications may include severe myocarditis (the most common cardiovascular adverse event, with a mortality rate of approximately 50%), myopericarditis, cardiac dysfunction/heart failure, arrhythmias, Takotsubo syndrome, or myocardial infarction. The mortality rate associated with cardiovascular toxicity is substantial. Adverse cardiovascular events typically manifest early in the treatment course and can occur after the first infusion, with most cases occurring within the first treatment cycles. However, later events have also been reported [2,9,11].

CART-cell therapy (Chimeric antigen receptor T cell) is an innovative approach in cancer treatment that involves modifying a patient’s own T-cells to target and destroy cancer cells. It is used for the treatment of acute lymphocytic leukemia and aggressive B-cell lymphomas.

CAR-T therapy may have cardiovascular toxicity, primarily due to cytokine release syndrome and immune-related adverse events.

Cardiac adverse event are LV dysfunction, HF, cardiac arrhythmias, pericardial effusion, Takotsubo syndrome and cardiac arrest [9,11].

Haematopoietic stem cell transplantation (HSCT) constitutes a potentially curative therapeutic option for many haematological malignancies.

During the early phase following HSCT, atrial fibrillation is the most commonly observed cardiovascular event, although other cardiovascular complications can include HF, arterial hypertension, hypotension, pericardial effusion, or thrombotic events. Long-term adverse effects encompass diabetes mellitus, abnormal lipid levels, metabolic syndrome, arterial hypertension, HF, coronary artery disease, conduction abnormalities, and pericardial effusion.

Acute graft versus host disease (GVHD) is linked to thrombosis and inflammatory damage to the heart muscle (myocarditis), HF, conduction irregularities, arrhythmias, and pericardial effusions. Meanwhile, chronic GVHD has been associated with an elevated risk of arterial hypertension, diabetes mellitus, and abnormal lipid levels [9].

Alkylating Agents (Cyclophosphamide, Isophosphamide/Cisplatin, Oxaliplatin, Carboplatin (containing platinum)) establish connections with DNA, hindering its replication and prompting cell apoptosis. They find application in the treatment of various cancers, including testicular, breast, ovarian cancer, leukemia, lymphoma, multiple myeloma, and sarcomas.

Cisplatin, albeit infrequently, can lead to LV dysfunction and HF. This occurs because it necessitates a high intravenous volume to prevent renal toxicity, which may result in symptomatic HF in patients with pre-existing CV conditions. Cisplatin also heightens the risk of dyslipidemia, obesity, metabolic syndrome, and myocardial infarction in long-term survivors.

Platinum-containing chemotherapy carries the potential for vascular complications, including vasospasms, myocardial infarctions, and both venous and arterial thrombosis.

Microtubule inhibitors (taxans, e.g., Paclitaxel and Docetaxel) are used to treat ovarian, breast, non-small cell lung, and prostate cancers.

Their antineoplastic effect involves stabilizing guanosine diphosphate-bound microtubules, thus inhibiting cell division.

Cardiovascular toxicity arises from the reduction in calcium amplitude in cardiomyocytes during contraction, diminishing cardiomyocyte contractility and resulting in LV dysfunction and HF. Microtubule inhibitors are also linked to an increased risk of stable angina and coronary artery disease (with a cardiac ischemia incidence of 1–5%) [9,11].

Radiotherapy (RT) is a valuable treatment modality used in a variety of cancer types, including breast, lung, esophagus, thyroid, prostate, mediastinal lymphoma, and head and neck tumors, often as part of a comprehensive therapeutic approach.

Its efficacy lies in disrupting DNA, causing local inflammation, and promoting tissue fibrosis.

However, RT can have detrimental cardiovascular consequences. Repeated episodes of ischemia due to inflammation can trigger microvascular endothelial cells in the pericardium to undergo fibrosis. RT also induces intravascular inflammation in epicardial coronary arteries, reducing oxygen and nutrient supply to the myocardium, leading to cardiomyocyte necrosis and fibrosis. Additionally, it contributes to a prothrombotic state, vasospasm, increased recruitment of monocytes and macrophages to the tunica intima, and greater inflow of lipoproteins, all of which accelerate existing atherosclerosis or initiate new atherosclerotic plaques.

Vasculopathy typically presents as severe, diffuse, long, and concentric coronary lesions, often asymptomatic due to reduced pain sensation [2,17].

Cardiotoxic effects encompass pericarditis, CAD, peripheral artery disease (PAD), valvular heart disease (notably aortic stenosis and mitral regurgitation), LV dysfunction, HFpEF, arterial hypertension, and occasionally, arrhythmias [2,17,18].

These effects may emerge years after completing therapy. The risk of cardiotoxicity is higher in cases involving a higher radiation dose (≥30 Gy), combined therapy with anthracyclines, heart irradiation, younger age (<25 years), prior cardiovascular dysfunction, and the presence of cardiovascular risk factors.

CAD risk increases within a decade postRT, with left-sided breast cancer patients, in particular, facing a heightened risk of CAD, primarily in the distal part of the left anterior descending (LAD) coronary artery and diagonal artery, accounting for the majority of lesions [17,19].

Figure 1 summarizes the different classes of cancer therapy and their cardiovascular effects undergoing imaging assessment.

## 4. Serum Cardiac Biomarkers to Detect Cardiotoxicity

The literature regarding the use of serum cardiac biomarkers for assessment of the risk of CTR-CVT before cancer therapy is limited and recommendations are primarily based on expert opinions. The 2022 ESC guidelines suggest that measurement of serum cardiac biomarkers, including cardiac troponin (cTn) I or T and natriuretic peptides (NPs) (e.g., B-type natriuretic peptide [BNP] or N-terminal pro-BNP [NT-proBNP]) can help assess the baseline CV risk in patients undergoing various cancer therapies (including anthracyclines, HER2-targeted therapies, VEGF inhibitors, proteasome inhibitors, ICI, CAR-T, and tumor-infiltrating lymphocyte therapies). This assessment can aid in identifying individuals who may benefit from cardioprotective therapy.

Baseline measurement of NPc and/or cTn is recommended in all patients with cancer at risk for CTRCD if the degree of change in biomarkers is to be used to detect CTRCD during treatment [9].

However, in cancer patients and those undergoing cancer treatment, there are no universally defined cutoff values for cardiovascular biomarkers. In addition, NP and cTn levels may vary between laboratories and are influenced by factors such as age, sex, renal function, obesity, infections, and other health conditions such as atrial fibrillation and pulmonary embolism [9].

Thus, any increase in biomarker levels must be interpreted by considering the patient’s clinical situation, including the time of cancer treatment and any comorbidities [9].

However, a clear association between anthracycline-associated troponin increase and future development of LV dysfunction has been shown. A 2020 meta-analysis studied the utility of cTn and/or NPs in 61 studies with 5691 patients. Patients with cTn elevation after anthracycline treatment had a seven-fold increase in risk of LV dysfunction, with a sensitivity of 54%, specificity of 79%, and a very high negative predictive value of 93%. Mean BNP/NT-proBNP levels also increased in patients after treatment, but the available evidence did not consistently indicate prediction of LV dysfunction during chemotherapy [20]. Cardioprotective therapy with beta blockers and ACE inhibitors/ARBs to mitigate cardiotoxicity during cancer therapy was associated with a decrease in serum cTn [20].

The 2022 ESC guidelines for patients on anthracycline therapy recommend measuring NP and cTn in subjects at high and very high risk of CTRCD at various times, including at baseline, before each course of treatment, and at 3 and 12 months after the end of therapy. Low-risk subjects can be monitored at baseline, potentially every other cycle during treatment, and possibly at 3 months after completion of therapy [9].

Emerging biomarkers such as myeloperoxidase and micro-RNAs show promise for risk stratification of cardiotoxicity before cancer treatment [9].

Biomarkers may help identify early signs of CTRCD, allowing for timely intervention. However, more research is essential to ascertain the best biomarkers and monitoring approaches.

## 5. Imaging Modalities to Detect Cardiotoxicity

CV imaging plays a crucial role in identifying patients with subtle CV dysfunction, evaluating pre-existing cardiac conditions before initiating cancer therapy, and serving as a reference for tracking changes during treatment and long-term follow-up [9].

Transthoracic echocardiography stands as the preferred imaging modality for baseline risk assessment. It offers quantitative evaluations of LV and right ventricle (RV)function, chamber size, LV hypertrophy, regional wall motion abnormalities, diastolic function, valvular heart diseases, pulmonary arterial pressure (PAP), pericardial diseases and LV GLS. These assessments are essential for influencing therapeutic decisions [9,21].

When echocardiography is not available or proves inconclusive, *CMR* is a valuable alternative for assessing cardiac function [9]. CMR is the gold standard for measuring LV and RV volumes and systolic function, offering insights into tissue characterization, blood flow, and perfusion. Cine imaging employs steady-state free precession sequences, providing excellent contrast-to-noise ratios among cardiac structures and allowing assessment of the motion of each cardiac component.

CMRm encompasses pre-contrast T1-weighted (T1w) and T2-weighted (T2w) images to detect regions with increased signal intensity, indicative of fat and tissue edema, respectively. Late gadolinium enhancement (LGE) imaging facilitates the visualization and quantification of myocardial reparative fibrosis and scar tissue. Modern mapping techniques enable precise quantification of myocardial composition by measuring T1 and T2 relaxation times.

Native (pre-contrast) T1 times reflect changes in myocardial extracellular water content, focal or diffuse fibrosis, and fat and amyloid content. Myocardial extracellular volume (ECV) quantification, derived from pre- and post-contrast myocardial and blood T1 values and corrected for blood hematocrit, serves as an indicator of diffuse interstitial fibrosis, provided other potential causes of increased extracellular space have been ruled out. T2 mapping is particularly useful for detecting edematous myocardium. Phase-contrast imaging aids in assessing the severity of valve regurgitation or stenosis [22,23,24,25].

Table 1 summarizes the potential scanning sequences that can be included in a tailored CMR protocol for evaluating cardiotoxicity.

Multigated acquisition nuclear imaging (MUGA) can be considered as a third-line option when both echocardiography and CMR are unavailable for assessing LVEF. However, it should be used sparingly due to radiation exposure and its limitations in providing comprehensive information [9]. CCT also enables the evaluation of heart chamber function. Retrospective ECG-gated CCT can offer volumetric and morphological data for both ventricles and may serve as a reliable and accurate alternative to CMR [26,27]. Additionally, for patients with a low to intermediate pre-test probability of CAD, CCT is a robust imaging modality with high sensitivity for ruling out obstructive CAD [28,29].

Functional imaging tests for myocardial ischemia, including stress echocardiography, CMR, andnuclear myocardial perfusion imaging (using physical or pharmacological stress depending on the patient’s clinical status), are recommended for diagnosing the presence and extent of myocardial ischemia and assessing the need for therapeutic interventions [7,9].

Table 2 provides an overview of the relative strengths and limitations of cardiovascular imaging techniques.

## 6. Multimodality Imaging Assessment of the Different Cardiovascular Clinical Conditions Related to Cancer Therapy

### 6.1. Baseline Assessment and CTRCD (LV Dysfunction/HF)

Before initiating cardiotoxic cancer therapy, it is crucial to perform a baseline assessment of LV function to identify any underlying cardiac dysfunction within the 3 months preceding treatment initiation.

Echocardiography remains the primary tool for detecting LV dysfunction, with a focus on accurately measuring LVEF. Three-dimensional (3D) echocardiography is the preferred method for assessing LVEF and cardiac volumes due to its enhanced accuracy and lower inter-observer variability (approximately 5–6%) [9,30]. When 3D echocardiography is not feasible, such as when it is unavailable or has poor tracking, the modified two-dimensional (2D) Simpson’s biplane method is recommended [9,31]. In cases where echocardiography image quality is inadequate, the use of ultrasound-enhancing contrast agents can improve the evaluation of LV function and volumes [32].

GLS, assessed through speckle tracking using three apical views, is a more sensitive and reproducible measure of LV systolic function than LVEF. It can detect subclinical cardiac dysfunction before abnormalities in LVEF become apparent. GLS is a valuable screening tool for risk stratification before cancer treatment and should be evaluated in all patients requiring baseline echocardiograms [33,34,35]. GLS values can be interpreted as follows: GLS <−16% as abnormal, GLS >−18% as normal, and GLS values between −16% and −18% as borderline [33]. Both LVEF and GLS are used to detect cardiotoxicity during and after anticancer treatment. A reduction in GLS of≥15% from the baseline is considered suggestive of subclinical cardiotoxicity and potential future LV dysfunction. It is important to report relative changes in GLS compared to previous measurements [33,36].

The SUCCOUR (Strain Surveillance of Chemotherapy for Improving Cardiovascular Outcomes) trial showed that initiating treatment with ACEI or ARBs and betablockers in patients undergoing cardiotoxic cancer therapy, based on a ≥12% relative reduction in LV GLS at any follow-up time point, results in a smaller decrease in LVEF compared to treatment after a decline in LVEF has already occurred [11,37].

Figure 2 shows an example of subclinical CTRCD.

However, it is important to note that there is some inter-vendor variability in GLS measurements of up to 3.7%, and GLS can be influenced by changes in loading conditions that commonly occur during chemotherapy, such as volume fluctuations due to intravenous fluids, vomiting, or diarrhea. Therefore, measuring systemic arterial blood pressure is recommended during resting echocardiography and should be included in the echocardiography report [9,38].

Myocardial work, which considers myocardial deformation and afterload, offers a more accurate assessment of cardiac function [39]. Indices of myocardial work, such as global work index (GWI), global constructive work (GCW), and global work efficiency (GWE), appear to be early and sensitive markers of progression towards cardiotoxicity in patients undergoing anthracycline therapy with or without trastuzumab [40,41,42]. There is also emerging evidence suggesting that myocardial work reflects myocardial function and energy utilization in patients receiving PD-1 inhibitor treatment for lung cancer [43]. However, further research is needed to establish the routine clinical use of myocardial work in managing cardiotoxicity.

While RV function has not traditionally been a part of the definition of cardiotoxicity, recent evidence indicates that RV abnormalities are prognostically significant in patients undergoing anthracycline and trastuzumab therapy [44]. RV free wall longitudinal strain appears to be an early marker for subclinical RV toxicity, although further validation is needed in clinical trials with longer follow-up periods [45,46].

Several studies have demonstrated progressive impairment of diastolic function in patients treated with anthracyclines. Although it may precede systolic dysfunction, the prognostic value of diastolic dysfunction is modest [47].

Left atrial strain, specifically reservoir and conduit reduction, frequently occurs early during anthracycline therapy and correlates significantly with routine echocardiographic diastolic parameters. This suggests a potential role in early cardiotoxicity detection [48].

In patients receiving anthracyclines, a decrease in peak mitral annular systolic velocity (S’) has been observed early and persists for several years after treatment, indicating its potential as an early marker of cardiotoxicity [49].

An increase in the Tei myocardial performance index, calculated as the sum of isovolumetric contraction and relaxation time divided by the ejection time [50], has been noted in patients undergoing anthracycline treatment. Changes in the Tei index are detectable earlier than alterations in the E/A ratio. Nevertheless, it does not exhibit a clear correlation with the decline in LVEF, and its long-term clinical relevance remains uncertain [51,52].

Dobutamine stress echocardiography has shown promise in identifying high-risk patients for cardiac dysfunction induced by chemotherapy drugs by detecting those with reduced LV contractile reserve [53,54]. However, its use in the early management of cardiotoxicity is limited by a lack of confirmed data in the literature [55,56].

Non-invasive assessment of coronary flow reserve (CFR) in the LAD coronary artery during dipyridamole stress echocardiography could be a valuable tool in patients treated with anticancer drugs. Impaired CFR has been observed in patients receiving sunitinib treatment, and it correlates inversely with treatment duration and inflammation markers, suggesting that inflammation may contribute to cardiac dysfunction related to sunitinib [56,57].

If transthoracic echocardiography yields poor image quality, CMR is acknowledged as a screening method for chemotherapy-related cardiotoxicity due to its precision, reproducibility, and capacity to detect subtle changes in RV and LV function [9]. The temporal variability in LVEF measurement by CMR is low, estimated at about 2.4–7.3% [58]. Functional changes in post-chemotherapy RV are linked to a noticeable decline in LVEF, especially in patients receiving combined therapy with trastuzumab [59].

More recently, CMR-derived GLS, using feature tracking (FT), has demonstrated its ability to detect LV dysfunction before a decline in LVEF and to independently predict all-cause mortality across various cardiomyopathies [60]. Reductions in both global circumferential and longitudinal strain have been observed in patients undergoing doxorubicin and trastuzumab treatment, and these changes correlated with subclinical declines in LVEF [61,62], suggesting its potential utility in monitoring early cardiotoxicity from chemotherapy. While assessing myocardial deformation using FT-CMR is feasible, it has greater temporal variability than echo-derived GLS, and evidence supporting FT-CMR-directed clinical management to improve CTRCD outcomes is currently lacking [63].

CMR measurement of LV mass also decreases following anthracycline administration, offering an additional biomarker for cardiotoxicity. This reduction in LV mass during treatment is attributed to at least a 40% reduction in cardiomyocyte size, partly offset by increases in ECV [64].

Some small-scale studies employing CMR have also demonstrated early myocardial edema following anthracycline therapy using T2w sequences [65]. The presence of edema has been associated with persistent RV function reduction in follow-up examinations [6]. Animal studies have shown increased cardiac edema with subsequent development of myocardial fibrosis after anthracycline exposure [66].

Although fibrosis is a known consequence of anthracycline treatment, LGE is not commonly observed and is not associated with outcomes, likely due to the diffuse nature of anthracycline-mediated fibrosis, which may be interstitial and therefore not well visualized with LGE-based methods [67,68].

Diffuse myocardial fibrosis induced by anthracycline therapy can manifest several years after completing treatment and may be assessed using T1 mapping [69,70]. However, an early decrease in T1 value after initial anthracycline treatment can predict the development of anthracycline cardiomyopathy after chemotherapy completion [71].

CMR appears capable of detecting early inflammatory involvement (elevated native T1 and T2) and interstitial fibrosis and remodeling (elevated native T1 but not T2) [21,25].

The ECV also increases in patients after anthracycline therapy compared to healthy controls and may rise within the first few months after treatment [66,71]. This increase in ECV is not solely due to expansion of the interstitial space; cardiomyocyte loss and atrophy may also contribute to its elevation [71,72]. Clinically, an increase in ECV has been associated with diastolic dysfunction, larger atrial volumes, and elevated short-term mortality [67]. However, further research is needed to validate the clinical use of ECV.

Historically, MUGA has been used to assess LV systolic function due to its availability, accuracy, and reproducibility [9]. Currently, MUGA is considered the third-line imaging choice for systolic function assessment in cardio-oncology due to excessive radiation exposure and its inability to provide information about RV function, atrial sizes, or valvular and pericardial diseases. MUGA may be considered in specific scenarios, such as patients with suboptimal echocardiographic windows, patients with CMR-incompatible implanted devices (e.g., patients after mastectomy with tissue expanders), and in centers without access to CMR [9,11].

Retrospective multiphase ECG-gated CCT can offer detailed morphological and functional information for both ventricles and may be considered an accurate and reproducible alternative to CMR [26,27]. Furthermore, CCT-derived ECV appears to be a potential biomarker for dynamically monitoring anthracycline cardiotoxicity in breast cancer patients [73,74].

It is clear that LVEF is an insufficiently sensitive parameter for prediction of HF and is incapable of detecting early dysfunction of myocardial contraction. The use of new parameters, such as GLS or myocardial work by speckle-tracking echocardiography or the evidence and quantification of myocardial edema and/or fibrosis by CMR helps us to recognize myocardial dysfunction early and to differentiate acute and late cardiotoxicity due to cancer therapy.

Table 3 summarizes echo, CMR, MUGA, and CCT parameters discussed in the text that are useful for identification of CTRCD, identifying those that alter early and those that alter later.

The downward arrow indicates a decrease in parameter value indicative of CTRCD, while the upwardarrow indicates an increase in parameter value indicative of CTRCD.

“Early” indicates that the parameter may alter early in CTRCD, while “later” indicates that the parameter alters later in CTRCD.

### 6.2. Myocarditis

Myocarditis is typically associated with ICIs, which have recently become a part of the treatment regimen for resistant malignancies [2]. Diagnosing ICI myocarditis can be challenging with routine echocardiography alone. In patients with myocarditis, a reduction in GLS as detected by echocardiography at the time of diagnosis has been observed and is linked to poorer outcomes [75].

When there is suspicion of ICI myocarditis, CMR imaging, including T1 and T2 mapping, and LGE, is recommended to aid in clinical decision-making. This includes determining whether to discontinue immunotherapy, the necessity of performing an endomyocardial biopsy and guiding the administration of high-dose steroids. The diagnosis of myocarditis on CMR is established using the updated Lake Louise criteria [22,76]. Endomyocardial biopsy is indicated in cases of an ongoing unstable hemodynamic state, uncertain diagnosis, or limited access to CMR [11].

In situations where CMR is not available, fluorodeoxyglucose (FDG)–positron emission tomography (PET) imaging may also be considered for diagnosing myocarditis [77].

CCT can be useful for ruling out obstructive coronary CAD when myocarditis is suspected [29].

For individuals with ICI myocarditis, follow-up CMR at 3 to 6 months has prognostic value [22,76].

### 6.3. Coronary Artery Disease/Myocardial Ischemia

Functional tests for detecting myocardial ischemia, such as stress echocardiography, perfusion CMR, or nuclear myocardial perfusion imaging, should be conducted in symptomatic patients (those with stable angina or limiting dyspnea) when clinical suspicion of CAD is present. This is particularly important before initiating cancer therapies associated with vascular toxicity or RT [9].

CCT can be employed to assess coronary atherosclerosis prior to commencing cancer therapy, aiding in the identification of the patient’s risk category for subsequent clinical management. For individuals undergoing chemotherapy, CCT serves to rule out acute coronary syndrome resulting from vasospasm, accelerated atherosclerosis, or thrombosis [29]. CCT can also identify hemodynamically significant coronary lesions by estimating fractional flow reserve using FFR-CT, an emerging non-invasive tool that provides valuable prognostic information and enables the optimization of therapeutic strategies [28,29]. Furthermore, CCT allows for the evaluation of heart chamber morphology, function, and tissue vitality thanks to the enhanced spatio-temporal resolution of modern-generation scanners.

In the post-radiation/chemotherapy population, single photon emission computed tomography (SPECT) myocardial perfusion imaging (MPI) plays a prognostic role in detecting obstructive CAD. This nuclear scan employs a pharmacological stress protocol with dipyridamole or dobutamine and TC-99m-tetrofosmin as a nuclear marker injected in two phases after pharmacological stress and later during a rest protocol. SPECT MPI does not require physical exercise, is not hindered by sonographic limitations, and can be performed immediately after radiation therapy.

Patients with extensive CAD risk factors undergoing high-risk oncologic surgeries may benefit from traditional SPECT imaging for improved cardiac risk assessment.

Positron emission tomography (PET MPI) is considered as the gold-standard technique for diagnosing myocardial metabolism and assessing microvascular function due to its ability to evaluate ischemia [11,28].

### 6.4. Takotsubo Cardiomyopathy

Takotsubo cardiomyopathy is more common in patients with cancer and it is associated with numerous chemotherapy agents.

Echocardiography is usually the first test of choice, showing the specific wall motion abnormalities—typically (about 75%) with akinesis of the mid and apical LV segments and mid-ventricular (15–20%) and basal ballooning (1%)—and eventual RV involvement (which is a prognostic marker of unfavorable outcomes) [9,11].

CMR can be a helpful additional test to distinguish between other potential etiologies of cardiomyopathy, such as myocarditis. Typical CMR features include specific wall motion abnormalities and diffuse and extended myocardial oedema, in particular in the regions with akinesis and absence of significant irreversible tissue injury (LGE), although subtle fibrosis may be rarely seen [11,22].

### 6.5. Pericardial Disease

Acute pericarditis, potentially accompanied by significant effusion that could lead to conditions like tamponade, has been linked to various chemotherapy drugs and intensive RT [2,9,11]. CMR is pivotal in evaluating pericardial disease, offering insights into anatomy, functionality, and tissue characterization. CMR methods like T1w imaging, cine imaging, and LGE provide comprehensive data [78,79]. CCT can detect acute pericarditis by highlighting an enhanced, thickened pericardium and any accompanying effusion. Based on Hounsfield unit (HU) measurements, CCT can differentiate between fluid types in the pericardial space [78].

After undergoing RT, some patients might develop chronic pericardial issues, including constrictive pericarditis. CMR helps in identifying these conditions by showing signs like myocardial tagging, indicating parietal and visceral pericardium adhesion. The presence or absence of pericardial LGE can indicate whether the condition is inflammatory or fibrotic [22,78]. For diagnosing constrictive pericarditis, CCT offers supplemental information by identifying features like a thickened or calcified pericardium, septal flattening, or bi-atrial enlargement. Importantly, CCT excels at spotting pericardial calcification, a crucial sign of constrictive pericarditis [78].

### 6.6. Pulmonary Hypertension

Although rare, cancer therapy can lead to pulmonary hypertension, as observed with dasatinib (a tyrosine kinase inhibitor with a 5% prevalence of PAH), cyclophosphamide, and other alkylating agents [2,9].

Echocardiography is the initial imaging choice, with recommended reassessment every 3–6 months for patients receiving PAH associated therapy [9,11].

CMR is considered to be the gold standard for assessing right heart structure, function, tissue characterization (e.g., LGE in areas subjected to chronic ventricular overload, often in the RV insertion points of the interventricular septum), and vascular abnormalities [11,23].

CCT provides a 3D quantitative assessment of RV and can detect fatty infiltration when CMR is unavailable or unsuitable. CCT is also useful for ruling out underlying CAD and lung disease [11,24].

### 6.7. Arterial Hypertension

Arterial hypertension is a common, modifiable cardiovascular risk factor, and its presence, especially in combination with other comorbidities, is an independent prognostic factor for survival. VEGF inhibitors and TKI increase the risk of arterial hypertension and may destabilize previously controlled hypertension [2,9,11].

While echocardiography is the primary investigative tool, CMR can offer complementary insights into the cardiovascular consequences of hypertension, such as LV hypertrophy and myocardial fibrosis. CMR can also help exclude underlying secondary causes like renal artery stenosis and adrenal masses [11,22,25].

### 6.8. Valvular Heart Disease

Valvular heart disease is a recognized complication of mediastinal RT, potentially affecting up to 10% of patients. However, modern treatment regimens have significantly reduced this number. There is typically a latent interval of 10–20 years between radiation exposure and the development of clinically significant heart valve disease, notably aortic stenosis and mitral regurgitation.

Echocardiography is the primary diagnostic tool, offering both qualitative and quantitative assessment of stenotic and regurgitant valves.

CCT and CMR can also aid in valve evaluation. CCT is particularly useful for stenotic valve planimetry and the assessment of suspected endocarditis, especially in conjunction with hybrid imaging like PET-CT.

CMR can provide additional information, especially when echocardiographic measurements are inconclusive, allowing for the assessment of valve morphology, measurement of LV and RV volumes and function, and quantification of valvular flows and velocities using phase contrast sequences [9,11,22,25].

### 6.9. Heart Failure with Preserved EF (HFpEF)

The risk of HFpEF in breast cancer survivors increases with higher cardiac radiation exposure during RT. While HFpEF in breast cancer survivors is not extensively studied, some data suggest that coronary microvascular compromise may contribute to its pathophysiology [2,9,11,17,18,31].

Echocardiography serves as the initial imaging modality for diagnosing diastolic LV dysfunction, defining cardiac structure and function, and grading diastolic dysfunction based on LV filling patterns and filling pressures. In cases where diastolic dysfunction is suspected but filling pressures are normal or echocardiographic findings are inconclusive, stress echocardiography can provide additional insights.

CMR is valuable for uncovering underlying pathologies in HFpEF patients, and CMR-derived LGE and ECV measurements can enhance diagnostic and prognostic information [9,11,16,17,20].

Future research should focus on improving HFpEF prevention and treatment by gaining a deeper understanding of its etiology and identifying contributing risk factors. When breast cancer survivors develop HFpEF, treatment primarily involves initiating guideline-directed medical therapy and addressing underlying comorbidities [2,9,11,13,14,16].

First-line treatment for HFpEF includes SGLT2i, which have positive cardiovascular effects by promoting natriuresis, promoting diuresis, reducing blood pressure, and alleviating oxidative stress, inflammation, and myocardial fibrosis [13].

Two large randomized clinical trials involving 5988 and 6263 patients (EMPEROR-Preserved [80] and DELIVER [81], respectively) demonstrated the benefits of SGLT2i (empagliflozin [80] or dapagliflozin [81] at 10 mg orally once daily) compared to a placebo in patients with HFpEF and New York Heart Association HF class II to IV symptoms, structural heart disease or a recent HF hospitalization, elevated NPs level, and EF greater than 40%.

Despite slightly different primary endpoints (besides cardiovascular death and HF-related hospitalization, DELIVER included urgent HF visit) both studies demonstrated a reduction from 18% to 21% in the rate of HF hospitalization and cardiovascular death with SGLT2 inhibitors (HR: 0.79, 95% CI 0.69–0.90, *p* < 0.001 in EMPEROR-Preserved and HR: 0.82, 95% CI 0.73–0.92, *p* < 0.001 in DELIVER) [80,81].

SGLT2ihave been shown to have a favorable safety profile in HFpEF, with no increased risk of hypoglycemia compared to their use in diabetes management. However, careful attention should be given to potential side effects like urinary tract infections and volume depletion, particularly in high-risk patients [13,80,81].

The introduction of SGLT2i represents a significant advancement in HFpEF management. Their unique mechanisms of action, proven clinical effectiveness, and favorable safety profile make them an appealing treatment option for a condition that previously lacked effective therapies.

Ongoing research should continue to explore their full potential and refine their role in the comprehensive management of HFpEF.

## 7. Cardiac Arrhythmias Related to Cancer Therapy

### 7.1. Atrial Fibrillation

Cancer patients have a higher incidence of atrial fibrillation (AF), which can be attributed to various factors, including comorbid conditions like arterial hypertension and HF, older age (typically over 65 years), the direct effects of cancer (dehydration, altered sympathetic tone due to anxiety or pain, systemic inflammation), complications of cancer surgery (especially lung surgery), or as a side effect of certain anticancer drugs.

Numerous classes of cancer therapies, including anthracyclines, fluoropyrimidines, VEGF inhibitors, TKIs targeting BCR-ABL, Bruton TKIs, proteasome inhibitors, ICI, alkylating agents, microtubule inhibitors, CART-cell therapy, and immunomodulatory drugs, can induce AF [9].

Patients with both cancer and AF have an elevated risk of systemic thromboembolic events (stroke), major bleeding, intracranial hemorrhage, HF, and overall mortality [82].

The criteria for long-term anticoagulation in the general population, as determined by the CHA_2_DS_2_-VASc score (e.g., CHA_2_DS_2_-VASc score ≥2 for men, ≥3 for women as Class I, and score = 1 for men, =2 for women as Class IIa), also apply to cancer patients. However, cancer patients with AF may have a higher thromboembolic risk than those without cancer, potentially underestimating their risk. According to 2022 ESC guidelines, anticoagulant therapy may be considered for men with a CHA_2_DS_2_-VASc score of 0 and women with a score of 1 (Class IIb), after assessing bleeding risk using the HAS-BLED score [9,82].

In terms of stroke prevention, non-vitamin K antagonist oral anticoagulants (NOACs) are preferred over low-molecular-weight heparin (LMWH) and vitamin K inhibitors in cancer patients without a high risk of bleeding, severe kidney dysfunction, or significant drug interactions (Class IIa) [9]. It is crucial to consider potential contraindications or dose reductions of specific NOACs because of the risk of drug interactions with various chemotherapies.

### 7.2. Ventricular Arrhythmias

Ventricular arrhythmias (VAs) are relatively infrequent among cancer patients, but their occurrence rises in individuals with advanced cancer and CV comorbidities. They can result from the direct impact of cancer treatments on ventricular action potential channels or from chronic arrhythmic conditions triggered by cancer, inflammation, pre-existing cardiovascular conditions or new CTR-CVT [9].

The standard upper limit for QTc values is 450 ms for men and 460 ms for women. QTc intervals that extend beyond 500 ms are associated with a threefold higher risk of developing Torsades de Pointes (TdP). The 2022 ESC guidelines for cardio-oncology categorize risk factors for prolonged QTc and VAs into two groups: correctable and non-correctable. Correctable factors include medications that lengthen QTc (e.g., antiarrhythmics, antibiotics, antihistamines, antiemetics), bradycardia, and electrolyte imbalances caused by cancer treatments. Non-correctable factors include age over 65, female gender, family history of sudden cardiac death, congenital long QT syndrome (LQTS), and pre-existing renal or liver disease [9,82].

Cancer therapy-induced VAs are primarily associated with QTc prolongation, which can lead to TdP. The 2022 ESC guidelines offer a simple mnemonic, “AAGNO P-R-S-TV”, to recall ten high-risk drugs for QT interval prolongation (e.g., aclarubicin, arsenic trioxide, glasdegib, nilotinib, oxaliplatin, pazopanib, ribociclib, sunitinib, toremifen, vandetanib) [9,82,83].

A 12-lead ECG is recommended following any dose increase inQTc-prolonging cancer therapy. According to the current ESC guidelines, QTc interval changes exceeding 60 ms, as long as QTc remains below 500 ms, should not hinder cancer treatment. When calculating QTc in cancer patients, the Fridericia formula is recommended due to its lower margin of error in both high and low heartrate conditions [9].

Reversible causes of prolonged QT intervals should be addressed, with weekly ECG monitoring, VA risk assessment during treatment and consideration of alternative therapies. Management of anticancer therapy-induced VAs should align with general guidelines for VA management. Betablockers and class IB antiarrhythmics are suggested as the safest options, especially when the cancer drug is known to cause cancer therapy-related cardiac dysfunction. Amiodarone is favored for patients with structural heart disease or hemodynamic instability. Therapy decisions should be personalized, taking into account factors such as complication risk and projected life expectancy [9,82].

## 8. Long Term Follow-Up of Cancer Treatment Survivors

Assessment of CV risk at the conclusion of cancer therapy is crucial. Asymptomatic survivors showing new or persistent abnormalities at this stage are deemed at high risk for future CV events and require long-term monitoring.

Certain cancer treatments carry the highest risk of long-term CV toxicity, notably anthracycline and RT involving the heart. Progressive RT-related CV toxicity tends to manifest 5–10 years posttreatment, significantly increasing the risk of CAD and HF, with an incidence up to six times higher than in the general population.

Late CV complications are also observed in survivors who underwent HSCT, with the incidence of HF increasing up to 14.5% in women 15 years after HSCT. Risk factors for CV dysfunction postHSCT include age, anthracycline dose, chest radiation exposure, arterial hypertension, diabetes mellitus, and smoking.

The timing and frequency of cardiac surveillance depends on the risk for CTR-CVT as per the latest ESC guidelines. Ideally, the same imaging modality used for LV systolic function assessment before antineoplastic therapy should be used for ongoing monitoring.

A reasonable approach includes yearly or biannual imaging follow-up after therapy completion, especially for high-risk patients with CV risk factors or extensive cardiotoxic treatments.

The surveillance interval can be shortened if cardiac dysfunction is observed. In cases of metastatic disease and long-term treatment, the frequency of imaging surveillance may be reduced.

Currently, there is no recommendation for lifelong surveillance in survivors of trastuzumab and other targeted cancer therapies beyond 10 years unless other indications arise [9,11].

## 9. Future Directions in Cardiotoxicity Due to Cancer Therapy

Exploring new cardiovascular imaging markers to identify cardiotoxicity and deepen our comprehension of its pathophysiology is a vital research area. The PET/CT scan, offering both anatomical and metabolic details, stands out as a promising diagnostic tool. The glucose analogue ^18^F-FDG is predominantly used in oncological PET/CT scans. Under significant stress, heart cells might elevate their glycolytic metabolism. The level of myocardial^18^F-FDG uptake as assessed by the ^18^F-FDG PET/CT scan could potentially signal early cardiomyocyte changes that anticipate cardiac dysfunction, marking it as a potential tool in the early detection of cardiotoxicity [84].

Lastly, we think that by utilizing deep learning, artificial intelligence has the potential to streamline and enhance various aspects of imaging processes. It can automate mundane tasks, boost precision and consistency, and optimize efficiency across diverse imaging techniques. Furthermore, broader artificial intelligence applications extend to integrating clinical and laboratory parameters with multi-mode imaging data, aiding in the characterization of individual phenotypes and the improved identification and prediction of cardiovascular risks and potential treatment responses in cardio-oncology patients.

## 10. Conclusions

The interplay between cancer treatment and the cardiovascular system is complex. A multidisciplinary approach using advanced multimodality imaging offers immense potential in pre-treatment risk stratification, early detection of cardiovascular injury during therapy and monitoring the long-term cardiovascular effects posttherapy.

The newly introduced terminologies and classifications from renowned cardiology institutions emphasize the importance of understanding and classifying cardiovascular toxicity due to cancer treatment.

With different classes of cancer therapies presenting various cardiovascular effects, a comprehensive grasp of their dose- and time-dependent impacts is crucial. Echo, CMR, MUGA, and CCT have emerged as significant modalities to detect subtle cardiovascular dysfunctions, assess pre-existing conditions before commencing cancer therapy, and longitudinally track changes during and posttreatment.

We accentuate the importance of a multidisciplinary approach for risk-benefit discussions before starting potentially cardiotoxic treatments, especially for high-risk patients. Furthermore, continuous cardiovascular monitoring tailored to individual treatment and patient-specific risks is underscored.

While the current imaging techniques provide profound insights, the exploration of novel imaging markers like the PET/CT scan points towards the promising future directions in the realm of cardio-oncology.

## Figures and Tables

**Figure 1 life-13-02103-f001:**
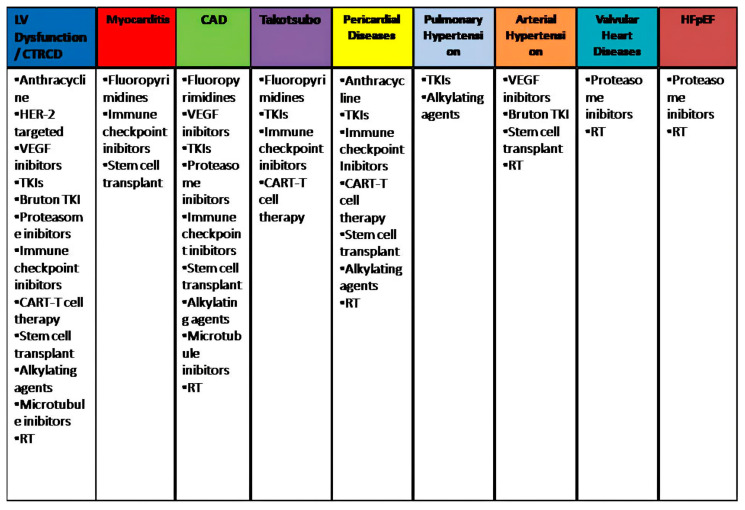
Summary of the different classes of cancer therapy and their cardiovascular effects undergoing imaging assessment, highlighted in different colors. CAD, coronary artery disease; CAR, chimeric antigen receptor; CTR-CD, cancer therapy-related cardiac dysfunction; HFpEF, heart failure with preserved ejection fraction; RT, radiotherapy; TKI, tyrosine kinase inhibitors.

**Figure 2 life-13-02103-f002:**
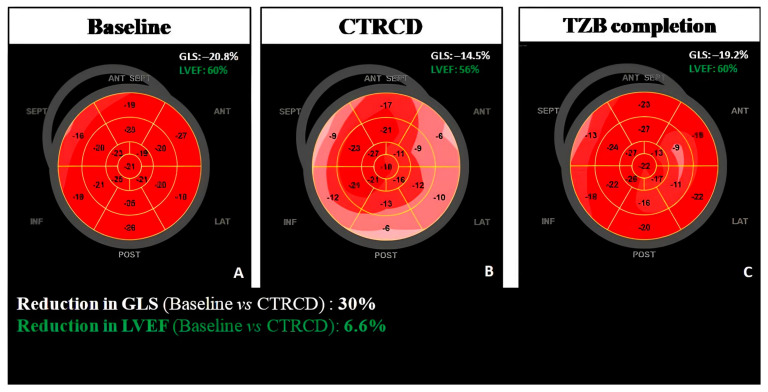
Clinical case of a patient developing subclinical CTRCD in an XI trastuzumab (TRZ) cycle but completing successfully TRZ, thanks to the timely cardioprotective therapy with recovery of both GLS and LVEF. Bull’s eyes of GLS at baseline (**A**), at the time of subclinical CTRCD (**B**) and at cancer treatment completion (**C**). CTRCD, cancer therapy-related cardiotoxicity; EF, ejection fraction; GLS, global longitudinal strain; TRZ, trastuzumab.

**Table 1 life-13-02103-t001:** Cardiac Magnetic Resonance Protocol for Cardiotoxicity.

Technique	Information
CINE (b-SSFP)	LV and RV volume and mass LV and RV function
Phase Contrast (when indicated)	Valvular heart disease severity
T1w ± fat saturation (when indicated)	Fat infiltration Pericardial thickness Anatomical info
T2w STIR	Myocardial oedema
Perfusion CMR (when indicated)	Myocardial perfusion defect
EGE (when indicated)	Thrombus Myocardial inflammation: hyperemia, capillary leak Ventricular myocardial microvascular obstruction
LGE	Ventricular myocardial reparative fibrosis Pericardial active inflammation
Native T1Mapping	Myocardial oedema Diffuse myocardial fibrosis
T2Mapping	Myocardial oedema
T1Mapping post-contrast	Diffuse myocardial fibrosis (ECV)
Tagging Technique/Feature tracking	Strain and strain rate analysis Parietal and visceral pericardium adhesion
Real-time freebreathing (when indicated)	Constriction

SSFP, balanced-steady-state free precession; ECV, extracellular volume; EGE, early gadolinium enhancement; LGE, late gadolinium enhancement; LV, left ventricle; RV, right ventricle.

**Table 2 life-13-02103-t002:** Summary of the relative strengths and limitations of the cardiovascular imaging modalities.

	Echo	CMR	CCT	MUGA/SPECT
Cost	Low	High	Medium	Medium
Radiation risk	-	-	+++	++++
Temporal resolution	++++	+++	++	
Spatial resolution	++	+++	++++	+
Coronary artery imaging	+	++	+++	-
Clinical Application	- can be used as first-line imaging test; - allows the assessment of numerous aspects of cardiac structure and function at rest and during exercise; - 3D-echo LVEF is accurate and reproducible; - GLS may identify subclinical cardiotoxicity or LV dysfunction from other causes.	- is the gold standard in evaluating heart structure and function; - allows tissue characterization of LV and pericardium; - ruling out underlying CAD.	- allows evaluation of heart structure and function when CMR is unavailable or unsuitable; - ruling out underlying CAD and lung disease (interstitial, COPD, CTEPH, cancer).	- MUGA: allows accurate and reproducible assessment of LVEF; - SPECT: allows assessment of myocardial ischemia and viability in underlying suspected CAD.
Limitations	- highly operator dependent; - need for GLS evaluation on the same vendor machine and ideally by the same operator; - inadequate imaging window; - limited evaluation of right ventricle.	- safety in patients with ferromagnetic implants; - use of gadolinium in patients with severe chronic renal failure; - breath holding; - no portability; - higher cost and lower available ; - claustrophobia.	- radiation exposure; - use of iodinated contrast; - no portability; - higher cost; - claustrophobia; - severe calcification may limit the assessment of severity of coronary lesion	- radiation exposure; - no portability; - MUGA: no information about other cardiac structures; - SPECT: low spatial resolution.

CAD, coronary artery disease; CMR, cardiovascular magnetic resonance; CCT, cardiac computed tomography; COPD, chronic obstructive pulmonary disease; CTEPH; chronic thromboembolic pulmonary hypertension; EF, ejection function; GLS, global longitudinal strain; LV, left ventricle; MUGA, multigated acquisition; SPECT, single photon emission computed tomography. The symbols“-/+” refer to the absence or presence of a specific feature of the cardiovascular imaging modality; each additional “+” sign indicates an increasing degree of the feature of the imaging modality (e.g., for radiation risk: “-”, no risk; “+”, low risk;“++”, mild to moderate risk;“+++”, moderate risk;“++++”, very high risk).

**Table 3 life-13-02103-t003:** Cardiovascular imaging modality parameters useful for identification of CTRCD.

Echo	CMR	MUGA	CCT
Clear evidence in the literature: - Early diagnosis: LV GLS ↓ - Later diagnosis: LV EF (3D) ↓ Emerging evidence in literature: - Myocardial Work: GWI, GWE ↓ GCW, GWW ↑ (early) - RV function: RV dimensions or areas ↓ RV S’ ↓ TAPSE ↓ PASP ↑ Free wall RV longitudinal strain ↓ (early) - LV diastolic function: Mitral E velocity E/A ratio e’ mitral velocity ↓ E/e’ average ratio ↑ - Left atrial strain: Reservoir ↓ Conduit ↓ (early) - Tei index ↑ (early) - Dobutamine stress echo: LV contractility reserve ↓ (early) - Dipyridamole stress eco: Coronary flow reserve in the LAD coronary ↓	- Early: LV GLS ↓ LV GCS ↓ RV EF ↓ LV ECV ↑ LV Mass ↓ - Later: LVEF ↓ - LV early inflammatory involvement: T1 native ↑ T2 ↑ - LV interstitial fibrosis and remodeling: T1 native ↑ T2 ↔	LV EF ↓ (later)	RV EF ↓ (early) LVECV ↑ (early) LV EF ↓ (later)

CMR, cardiovascular magnetic resonance; CCT, cardiac computed tomography; CTRCD, cancer therapy related to cardiac dysfunction; ECV, extracellular volume; EF, ejection fraction; GCS, global circumferential strain; GCW, global constructive work; GLS, global longitudinal strain; GWE, global work efficiency; GWI, global work index; GWW, global wasted work; LGE, late gadolinium enhancement; LV, left ventricle; MUGA, multigated acquisition; PASP, pulmonary arterial systolic pressure; RV, right ventricle.

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
