# Peer review of "Multimodality Cardiovascular Imaging of Cardiotoxicity Due to Cancer Therapy"

_life, 2023, doi:10.3390/life13102103_

Round 1

Reviewer 1 Report

The authors evaluated the role of advanced multimodality imaging in monitoring patients before, during and after cancer treatment.

I have the following concerns:

1. Please reorganize the abstract and include Introduction, Methods, Results and Conclusion sections. 

2. Please include the general state of knowledge on multimodality cardiovascular imaging in cardiotoxicity due to cancer treatment in the Introduction section. 

2. What are the methods of the study? What kind of studies were selected? What were the search methods?

3. Please put the selected RCTs in Table?

4. What was the possible bias in the included studies?

Minor editing of English language required

Reviewer 2 Report

Dear editors of Life,

I fully appreciate the opportunity to review a manuscript entitled “Multimodality Cardiovascular Imaging in Cardiotoxicity due to Cancer Therapy”. Cardiotoxicities of various cancer therapies are clinically important issue. The paper deals with this issue with in-depth review. I have some suggestions for authors.

#1. Patients undergoing chemotherapy often develop atrial fibrillation. Can authors discuss about chemotherapy-related atrial fibrillation? Whether chemotherapy can directly cause atrial fibrillation or is just due to deterioration of general medical condition; whether it is temporary or permanent diagnosis of atrial fibrillation; do we need to anticoagulated even under chemotherapy…

#2. In line 89 – 91, authors suggest RAAS inhibitors and beta blockers for primary prevention of chemotherapy induced cardiotoxicities. What about sodium-glucose co-transporter 2 inhibitor (SGLT2i)? This drug is rapidly emerging as a main therapy for both HFrEF and HFpEF. Can authors discuss about the current role and future perspective of this drug?

#3. In paragraph starting from line 608, I suggest to discuss about the role of SGLT2i, a only proven drug for HFpEF.

#4. Although this review paper mainly focuses on imaging modalities, brief discussion about N-terminal pro-brain natriuretic peptide (NT-pro-BNP) can be helpful. Advanced imaging modalities such as cardiac MRI or PET-CT are not available on many parts of the world. However, laboratory tests including BNP can be better available.

#5. Can authors discuss about lethal ventricular arrhythmias or conduction system damage related with chemotherapy? Is it permanent or temporary?

Round 2

Reviewer 1 Report

The authors preferred not to reply for the majority of my questions. Thus, I suggest that the manuscript is not appropriate to be published.

Minor editing of English language required

Author Response

We are sorry  for the misunderstanding but disagree with the reviewer for the reasons expressed above. Thank you.